# Tumor Area Highlighting Using T2WI, ADC Map, and DWI Sequence Fusion on bpMRI Images for Better Prostate Cancer Diagnosis

**DOI:** 10.3390/life13040910

**Published:** 2023-03-30

**Authors:** Rossy Vlăduț Teică, Mircea-Sebastian Șerbănescu, Lucian Mihai Florescu, Ioana Andreea Gheonea

**Affiliations:** 1Doctoral School, University of Medicine and Pharmacy of Craiova, 200349 Craiova, Romania; rossy.teica@gmail.com; 2Department of Medical Informatics and Biostatistics, University of Medicine and Pharmacy of Craiova, 200349 Craiova, Romania; 3Department of Radiology and Medical Imaging, University of Medicine and Pharmacy of Craiova, 200349 Craiova, Romania; lucian.florescu@umfcv.ro (L.M.F.);

**Keywords:** prostate cancer, bpMRI, PI-RADS, MRI fusion

## Abstract

Prostate cancer is the second most common cancer in men worldwide. The results obtained in magnetic resonance imaging examinations are used to decide the indication, type, and location of a prostate biopsy and contribute information about the characterization or aggressiveness of detected cancers, including tumor progression over time. This study proposes a method to highlight prostate lesions with a high and very high risk of being malignant by overlaying a T2-weighted image, apparent diffusion coefficient map, and diffusion-weighted image sequences using 204 pairs of slices from 80 examined patients. It was reviewed by two radiologists who segmented suspicious lesions and labeled them according to the prostate imaging-reporting and data system (PI-RADS) score. Both radiologists found the algorithm to be useful as a “first opinion”, and they gave an average score on the quality of the highlight of 9.2 and 9.3, with an agreement of 0.96.

## 1. Introduction

Prostate cancer is the second most common malignancy in men worldwide, causing approximately 350,000 deaths each year [1,2]. Magnetic resonance imaging (MRI) examination is used in the diagnostic pathway of prostate cancer to detect and localize the disease. Its results can be used to decide the indication, type, and location of a prostate biopsy to diagnose cancer. In addition, MRI has the potential to contribute information about the characterization or aggressiveness of detected cancers, including tumor progression over time [3,4].

The Prostate Imaging-Reporting and Data System (PI-RADS) score is the most commonly used system in prostate MRI examination and allows communication between physicians regarding the likelihood that a suspicious region contains clinically significant cancer. The scale from 1 to 5 is interpreted as follows: 1 and 2—low risk, 3—equivocal risk, 4—high, and 5—very high risk [5]. The PI-RADS system is based on elements identified during the MRI examination on a T2-weighted image (T2WI), apparent diffusion coefficient map (ADC), diffusion-weighted image (DWI), and dynamic contrast-enhanced (DCE) sequences [5,6,7].

T2WI is the basic pulse sequence on MRI because it can delineate the zonal anatomy and detect, localize, and stage prostate cancer, including extraprostatic extension and seminal vesicle invasion. Suspicious areas in the peripheral area of the prostate appear hypointense compared to the healthy glandular tissue. However, this aspect is non-specific, and benign entities such as atrophy, hemorrhage, prostatitis, and post-treatment changes can mimic cancer in this sequence. The transitional zone is often hypointense on T2WI, so cancer is more difficult to identify at this level. Suspicious lesions here are characterized by ill-defined edges, with an “erased charcoal” appearance, and sometimes they may have a spiculated or water-drop appearance. Prostate cancer can be invasive both inside and outside the prostate, where it typically invades the seminal vesicles or the neurovascular bundle, so the clear delineation of the prostatic capsule on this sequence is very important [7,8,9,10].

The ADC sequence is obtained by comparing the data obtained in the DWI sequences at different b values and represents the net displacement of water molecules on a timescale that reflects diffusion sensitization. Regions of interest are used to obtain measurements in suspicious focal areas in the prostate, and the low signal intensity confirms the restricted diffusion seen as hyperintense on DWI, which in both cases reflects the same underlying phenomenon [11]. The ADC sequences vary depending on the actual quality of the DWI sequences used and the b values of those that were used in the reconstruction. In this study, we used a conventional ADC map obtained from DWI images with low, intermediate, and high values (0, 800, and 1500) using the software integrated in the 3Dnet^TM^ platform provided by ©Biotronics3D, London, UK.

A DWI can analyze the property of water molecules in the body to be in Brownian motion. In simple fluids, the movement of water is free, but in tissues, it is restricted, especially by cell membranes, which, due to their hydrophobic character, act as a barrier that reduces the movement of intracellular and extracellular water. In an area with more cells (such as in cancer), the restricted diffusion of water molecules will demonstrate high signal intensity on DWIs with a high b value (800 or higher), thus increasing the visibility of tumor foci since higher b values offer a better contrast between the tumor and healthy tissue compared to b values lower than 800, which provide limited utility in this case. However, in MRI of the prostate, ultra-high b values of 1400 are used to provide suppression of benign prostate and focal prostatitis in the peripheral area and to improve prostate cancer (PCa) detection [12]. This can be challenging since increasing b values over 1000 s/mm^2^ favors image artifacts, leading to a lower-quality image. Therefore, when using ultra-high b values, it is necessary to use some practices that reduce the associated image artifacts, such as evacuation of the rectum before the examination. Images acquired with the DWI sequence contains very little anatomical information, and therefore, in practice, all DWI image findings are correlated with another sequence in order to better assess the loco-regional anatomy [13,14,15,16]. In this study we used DWI images with an ultra-high b value of 1500.

A dynamic contrast-enhanced sequence evaluates the neoangiogenic process found in cancer. An area becomes more suspicious with focal enhancement earlier than the adjacent parenchyma if it corresponds to suspicious areas on previous sequences.

According to PI-RADS, the role of DCE was reduced to the differentiation of PI-RADS 3 from PI-RADS 4 in the peripheral zone [5,17]. However, the DCE sequence or its reconstructions were not included in the fusion because it would have increased the noise level in the transition zone too much.

In the attempt to objectively characterize prostatic lesions, we must observe the differences between them, taking into account the anatomical area to which they belong. In the peripheral zone, the PI-RADS category of a lesion is primarily determined by the DWI/ADC sequences. A peripheral lesion becomes more suspicious when it demonstrates hypointensity in the ADC sequence and hyperintensity in the DWI sequence. A PI-RADS 3 lesion in the peripheral zone is assigned to PI-RADS 4 if the lesion demonstrates contrast uptake [5].

For the transition zone, T2W is the primary determinant sequence and is correlated with the DWI/ADC image findings in order to assign the correct PI-RADS assessment category. A PI-RADS 3 lesion becomes a PI-RADS 4 if the lesion demonstrates restricted diffusion (high signal intensity on the DWI sequence and low signal intensity on the ADC map) [5,7].

This study starts from the premise that the lower the signal intensity is on the T2WI and ADC map, and the higher the signal intensity is on the DWI sequence, the more suspicious that lesion is. Of course, a lesion requires additional characterization, such as the appearance of the contour, the size, and its contrast enhancement, but the ranking of the lesions is a great challenge in heterogeneous prostates. The aim of the current paper is to (1) obtain a method to highlight the possible malignant areas from MRI of the prostate and (2) discriminate between different PI-RADS patterns (3, 4, and 5) using just the fusion of T2WI, ADC, and DWI signals.

## 2. Materials and Methods

### 2.1. Dataset

The study was retrospective and included 80 patients with suspected PCa based on elevated PSA levels and/or abnormal digital rectal examination and in some patients with proven PCa for further staging. The dataset included examinations with a prostate image quality score (PI-QUAL) of 5 or 4, which corresponded to the PI-RADS V2.1 criteria, and both biparametric magnetic resonance imaging (bpMRI) and multiparametric magnetic resonance imaging (mpMRI) from the DCE sequence was ignored for labeling. All of the prostate MRI examinations were performed between December 2021 and December 2022 in the Medical Imaging Department of the University of Medicine and Pharmacy of Craiova using a Philips Ingenia 3.0T MRI scanner (Philips, Amsterdam, The Netherlands). The mean age of the patients was 69.34 (±8.28) years, ranging from 42 to 88 years. From each examination, the three sequences used in bpMRI were selected at the level of at least one common slice to be included in our fusion algorithm. A total of 204 resulting fusion images were used. All patient data were anonymized, and the study was approved by the Ethical Board of the University of Medicine and Pharmacy of Craiova (no. 3/12.01.2023).

### 2.2. MRI Protocol

From the performed pelvic sequences, our algorithm used only three, but all of them had to respect a series of common properties (Table 1).

All sequences used were axial obliques positioned perpendicular to the prostatic urethra and parallel to the base of the urinary bladder, with a field of view that was small (approximately 180 × 230) but large enough to cover the entire prostate and seminal vesicles. In order for the images to be superimposed, they must have slices of equal thickness (3 mm or less) and the same spatial gap, no greater than the thickness of the slices, otherwise there will be non-visible areas that will decrease the sensitivity of the examination.

Images had a larger field of view, were positioned perpendicular to the lumbar spine and parallel to the left and right iliac wing, had a slice thickness of at least 3 mm (but not greater than 6 mm), and had a spatial gap equal to or smaller than the slices.

The resulting images will have reduced effectiveness on the prostate, but they will have the potential to capture suspicious pelvic and extraprostatic lesions.

### 2.3. Algorithm

The proposed algorithm, developed in MATLAB (Mathworks, Natick, MA, USA) is composed of image resizing, inversions, color space changes, and different channel computations, and it ends up adding the ‘jet’ color map [18] to a grayscale intensity image.

The first step was to resize all images to 600 × 600, 8-bit, and grayscale, as they had different resolutions (Table 1). Thus, we obtained our original resized images for T2WI, ADC, and DWI, which were further used.

T2WI = uint8(imresize(T2WI, [600 600]));

ADC = uint8(imresize(ADC, [600 600]));

DWI = uint8(imresize(DW, [600 600]));

The second step was to compute the inversion of T2WI and ADC, thus obtaining the area of interest (tumor) represented by higher intensity values, as seen in Figure 1D,E. For this, we subtracted the actual image values from the maximum of 255.

T2WI = 255 − T2WI;

ADC = 255 − ADC;

**Figure 1 life-13-00910-f001:**
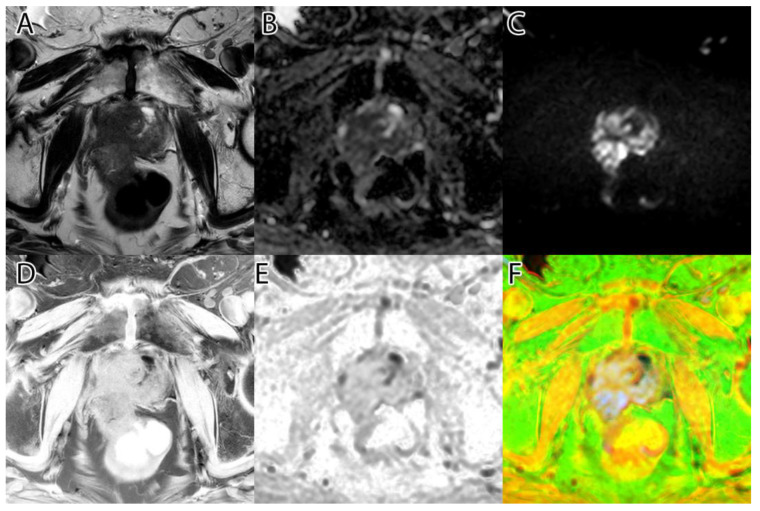
Sample image: (**A**)—T2WI; (**B**)—ADC; (**C**)—DWI; (**D**)—inverted T2WI; (**E**)—inverted DWI; (**F**)—RGB images resulting from image fusion of (**C**–**E**).

The third step was to combine the three images into an RGB (red, green, blue) image. T2WI was used for the red channel, ADC was used for the green, and DWI was used for the blue. The resulting image can be seen in Figure 1F.

RGB = cat(3, T2WI, ADC, DWI);

The fourth step was to convert the RGB color space to LAB color space [19], thus obtaining a luminance-based color space with two independent color channels: alpha and beta.

LAB = rgb2lab(RGB)

The fifth step was to obtain 8-bit grayscale images from the LAB color space channel components. For this, the luminance channel was multiplied by 2.55, as its values range from 0–100, while the other channels were left untouched.

luminance = squeeze(LAB(:,:,1));

luminance = uint8((2.55*luminance));

alpha = squeeze(LAB(:,:,2));

beta = squeeze(LAB(:,:,3));

The sixth and final step was to compute the final intensity mask by subtracting the square root from the square sum of the alpha–beta channels and subtracting the resulting value from the luminance. The resulting intensity image can be seen in Figure 2B and Figure 3B. By adding the ‘jet’ color map to it, we obtained a colored intensity map that highlights the tumoral areas, as seen in Figure 2C and Figure 3C.

combined-channels = uint8(sqrt(alpha .* alpha + beta .* beta));

intensity = luminance − combined-channels;

current-colormap = colormap(’jet’);

intensity-jet = ind2rgb(intensity, current-colormap).

**Figure 2 life-13-00910-f002:**
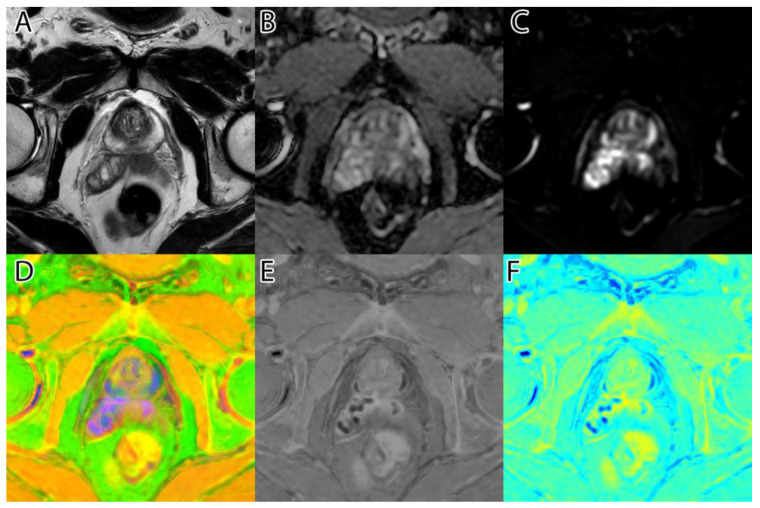
PI-RADS 2 patient: (**A**)—T2WI; (**B**)—ADC; (**C**)—DWI; (**D**)—RGB fused image; (**E**)—grayscale tumor intensity image; (**F**)—‘jet’-mapped tumor intensity image.

**Figure 3 life-13-00910-f003:**
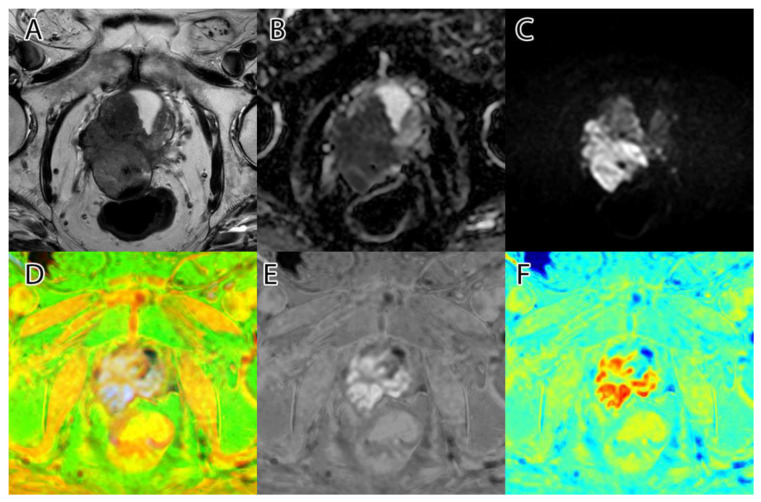
PI-RADS 5 patient: (**A**)—T2WI; (**B**)—ADC; (**C**)—DWI; (**D**)—RGB fused image; (**E**)—grayscale tumor intensity image; (**F**)—‘jet’-mapped tumor intensity image.

### 2.4. Ground Truth

Two trained radiologists segmented each image on the three bpMRI sequences and labeled them into four classes: PI-RADS 3, PI-RADS 4, PI-RADS 5, and extraprostatic tumor. For mpMRI examinations, the DCE sequence was ignored when labeling. There were no disagreements between the radiologists regarding the presence or labeling of the lesions, but there was subjectivity regarding the segmentation of the edge of the lesions pixel by pixel, both due to the human factor and the difference in experience between the radiologists.

Their overlapping segmentation of PI-RADS 3, PI-RADS 4, and PI-RADS 5 lesions were considered as ground truth, as seen in Figure 4.

Using the masks from the segmented images and imposing them on the intensity mask computed with the proposed algorithm, we computed the histogram of each segmented area (PI-RADS 3, PI-RADS 4, PI-RADS 5, and extraprostatic tumor), as seen in Figure 5.

### 2.5. Algorithm Evaluation

After visually inspecting Figure 5, we made several empirical cut points on the intensity histogram trying to split the intensities into the three classes of PI-RADS as follows:

PI-RAD 3—[100 164]

PI-RAD 4—[165 189]

PI-RAD 5—[190 220]

Using the resulting thresholds, each intensity image was binarized and compared to the ground truth label from the radiologists’ annotations.

### 2.6. Perceptual Evaluation

Both radiologists were asked to rank the quality of the tumoral highlight with marks ranging from 0 to 10, where 10 is the best highlight possible, and to express if they find that the algorithm could bring extra information for the diagnosis.

## 3. Results

Using the principles of the PI-RADS score, T2WI, ADC, and high b value DWI fusion managed to mark PI-RADS 4 and 5 with good accuracy in the 80 examinations chosen for the study because T2WI and DWI had the same slice thickness with no inter-slice gap and because the examinations did not include artifacts or movements of the patient in the area of interest.

The algorithm was used on all 204 images and computed the tumor intensity image. A sample from the original data, and the resulting fused image, can be seen in Figure 1.

Samples of the computed tumor intensity images can be seen in Figure 2 for a PI-RADS 2 patient and in Figure 3 for a PI-RADS 5 patient. Notice the difference in “jet” intensity between the control patient and the patient with extraprostatic invasion.

A sample of the ground truth marking can be seen in Figure 4.

The histogram of each PI-RADS label and extraprostatic invasion computed on all dataset images can be seen in Figure 5.

Figure 6 shows a collage of several cases and the resulting tumor intensity mask. The overlap between the PI-RADS class intervals and the original dataset can be seen in Figure 7. Using the empirically selected thresholds, we obtained 100 percent accuracy in identifying the PI-RADS 5 class on the images.

From Figure 8, we can understand why the fusion between T2 and DWI is insufficiently specific. By including the ADC map, our model was not affected by the T2 shine-through effect.

Another example of canceling the shine-through effect can be seen in Figure 9. An inexperienced examiner could be tricked by the fusion of T2WI and DWI, in which the left half of the peripheral zone appears to have diffusion restriction, easily noticeable as a lesion PI-RADS 5. Instead, the fusion including the ADC sequence shows a prostate without suspicious lesions. This is a classic example of chronic prostatitis.

Although our model is specific enough to differentiate lesions suspected of being PCa from inflammatory lesions, it has a reduced sensitivity to lesions with an equivocal risk of being PCa, especially those in the transitional zone. 

In the transitional zone, a lesion with a heterogeneous signal and obscured margins in the T2WI sequence, but not evident in the DWI and ADC sequences, is considered PI-RADS 3. Therefore, superimposing DWI and ADC over it only causes the addition of noise in these cases (Figure 10).

These aspects were foreseen and taken into account when working on the concept; therefore, the purpose of this model is to precisely mark the lesions with high and very high risk (PI-RADS 4 and PI-RADS 5) and extraprostatic invasion. Equivocal risk lesions (PI-RADS 3) could be detected by principles different from ours and are not the subject of this study.

Both radiologists found the algorithm to be useful as a “first opinion”, giving an average score on the quality of highlight of 9.2 and 9.3, with an agreement of 0.96. They both noted the extraprostatic non-tumoral highlights.

## 4. Discussion

The proposed algorithm highlights the tumoral areas of the prostate.

Fusion is a technique present in modern visualization and reporting platforms of medical images that allows the combining of two sequences and is most frequently used for combining positron emission tomography (PET) acquisitions with CT/MRI.

Being inspired by this technique, we noticed potential in combining other sequences within the same examination, but the limitation to only two sequences was not enough to obtain major results. On the other hand, merging too many sequences risks increasing the noise until the resulting image becomes redundant. Even the use of a subtraction reconstruction of the DCE experienced in the early stages of this paper proved to increase the noise of the obtained fusion too much. Experimentation with other sequences was not even attempted due to the desire to preserve the authenticity of the PI-RADS score. 

As seen in Figure 6, column D, when adding a ‘jet’ color map to the algorithm’s tumor intensity output, the highlighted zones appear in red and orange, while the less suspicious zones are colored in yellow. Non-tumoral surrounding tissue has colors ranging from green to blue.

Taking into consideration the PI-RADS 3/4/5 histogram channels in Figure 5, it can be seen that the PI-RADS 3 and 4 overlap and that PI-RADS 5 has higher intensities than the former. The similarities between PI-RADS 3 and 4 aspects are not new findings, as radiologists tend to add other information (besides intensity) such as positions and shapes for the diagnosis. 

Figure 5 also shows a wide variety of intensities at the level of the red histogram (PI-RADS 5) and the black histogram (extraprostatic tumor invasion) with the presence of a double-peak distribution at their level that is more present in the black histogram. The first peak, which signifies lower intensities, is more pronounced in both and can be argued by the lack of precision of the radiologists’ initial labeling at the edge of the lesions where there is a mix of pixels with high values representing the lesion and medium values representing the healthy adjacent tissue, especially since in some cases, the sequences are not perfectly aligned, thus increasing the possibility of including normal tissue in the tumor labels. Note that the first peak of the other histograms overlaps because in their case, the adjacent healthy tissue is prostatic. The black histogram, which shows a greater variety of intensities, is due to the adjacent healthy tissue, which in this case is extraprostatic tissue and has lower values than the adjacent healthy tissue from the rest of the categories. On the other hand, the second peak of the histogram looks similar to that of the red histogram, which can be explained by the fact that it is an extension of a high-grade, high-risk tumor that extended beyond the prostate.

The presence of the first peak in these histograms proves that manual segmentation, even by overlapping the segmentations of two experienced radiologists, is a limitation of this study.

As seen in Figure 6, image E2, the method is consistent and identifies high-risk lesions outside the prostatic area (see right side nodule in red (arrow)). The nodule was later confirmed through biopsy to be high-grade tumor prostate adenocarcinoma invasion.

With an average of highlight quality over 9/10 and with a high agreement between the radiologists that marked the highlight, the proposed method could find its way to clinical practice, offering, at least, an improvement in the reader’s confidence. Through the fusion of T2WI, ADC, and DWI, we obtained a method for highlighting suspicious lesions in the prostate. 

Several authors obtained satisfactory results by joining two sequences, but we did not find studies in the literature in which more were used. Nevertheless, they obtained higher diagnostic accuracies. Neves et al. (2022) [20] had a sample of 87 patients and revealed a statistically significantly higher capacity of fused images to evaluate the degree of myometrial invasion compared to standard MRI evaluation with T2WI, DWI, and DCE sequences. Mongula et al. (2019) [21] found that fusion of T2W with DWI showed a significant increase in diagnostic performance for the assessment of parametrial invasion in early-stage cervical carcinoma. Colvin et al. (2020) [22] reported that the fusion of DWI and T2WI increased sensitivity for the detection of prostate cancer complications. Brenner et al. (2012) [23] investigated the added value of fusion of high b value DWI and T2WI for the detection of pancreatic neuroendocrine tumors. Fusion imaging had a significantly better detection score, and a significantly better detection rate. Mir et al. (2010) [24] proved that the fusion of DWI with T2WI improves the detection of pelvic lymph nodes compared to T2WI alone.

Based on 27 other papers, Stec et al. (2018) [25] concluded that the perception and performance of radiologists decrease throughout the working day, making them more susceptible to errors. We consider that the use of the proposed method can significantly reduce the number of errors, especially after late hours. As we showed in Figure 9, our method remains valid even when there are varieties of the protocol because our method takes into account the principles behind the PI-RADS score. Having said that, we consider it to be specific to bpMRI and rarely influenced by the DCE sequence in mpMRI because its utility still remains a subject of discussion and is at least as good as the simple fusion between T2WI and DWI at the extraprostatic level [26]. Motivated by the advantages demonstrated by other researchers of two-sequence fusion and the high-quality result of our three-sequence fusion, we plan to extend this method to more segments examined through MRI, with a direct target being prostate cancer complications [22].

Our methodology does not imply any machine learning methodology, and it is not stochastic (the output results are dependent on the training data) [27]. In a review on diagnosis and localization of clinically significant prostate cancer on MRI images [28], the authors compared several deep-learning methodologies and note that all but one of the proposed methods provided strategies for visual inspection of the deep learning predictions (prediction heatmaps [29], gradient-weighted class activation mapping [30], and class-activation maps [31]). The results of our method can be visually inspected by comparing the sequences that were used, and thus, the risk of bias is not present. However, we have no intention of automated classification or segmentation; only highlighting is automated.

With the progress of machine learning methodology and artificial intelligence that we have seen in recent years [29,30,31,32,33], it is obvious that the progress in this field is greater than our relatively simple model. However, our model uses very few resources and can be easily integrated into clinical practice, and the images resulting from our methodology, in combination with other sequences, we believe could be used as input for machine learning algorithms.

## 5. Conclusions

This paper aimed to use bpMRI fusion to reduce the incidence of human errors caused by the large number of imaging examinations that radiologists report on a daily basis. The methodology presented illustrates an algorithm meant to highlight possible malignant areas on prostate MRI using the fusion of T2WI, ADC, and DWI signals and no machine learning algorithms. Since the involvement of the DCE sequence or its reconstructions would have significantly increased the noise level, it was not included in the fusion algorithm, and we can consider our model specific to bpMRI. However, this does not invalidate its principles when used in an mpMRI prostate examination.

Considering that the sequences used are internationally standardized together with the PI-RADS v2.1 criteria for classifying prostate lesions, this algorithm remains universally valid. Our fused sequence has proven useful in a first-opinion setting, accurately detecting PI-RADS 4 and 5 lesions. 

## Figures and Tables

**Figure 4 life-13-00910-f004:**
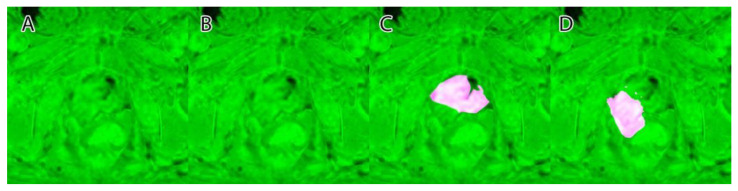
Ground truth: (**A**)—PI-RADS 3; (**B**)—PI-RADS 4; (**C**)—PI-RADS 5; (**D**)—extraprostatic invasion.

**Figure 5 life-13-00910-f005:**
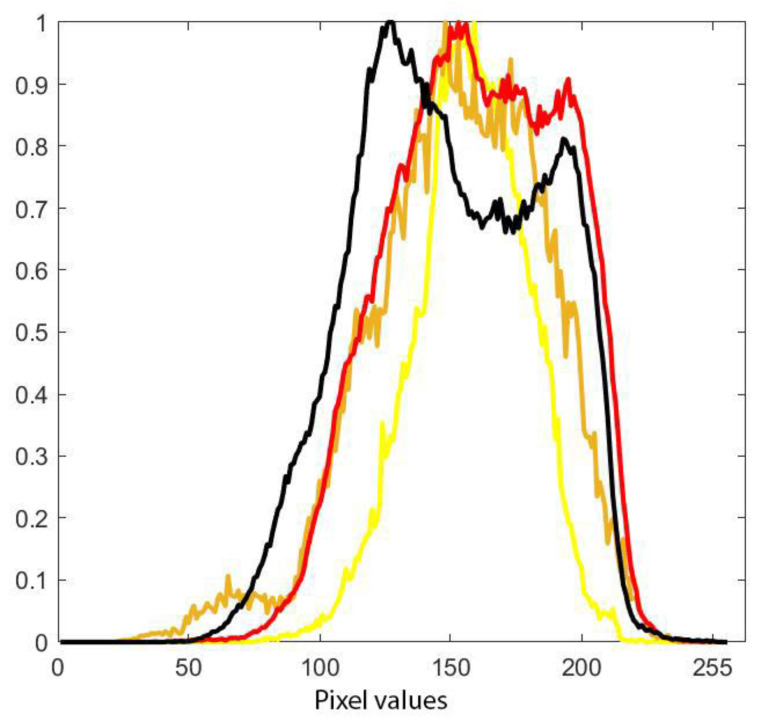
Pixel values on grayscale tumor intensity image: (**Yellow**)—PI-RADS 3; (**Orange**)—PI-RADS 4; (**Red**)—PI-RADS 5; (**Black**)—extraprostatic invasion.

**Figure 6 life-13-00910-f006:**
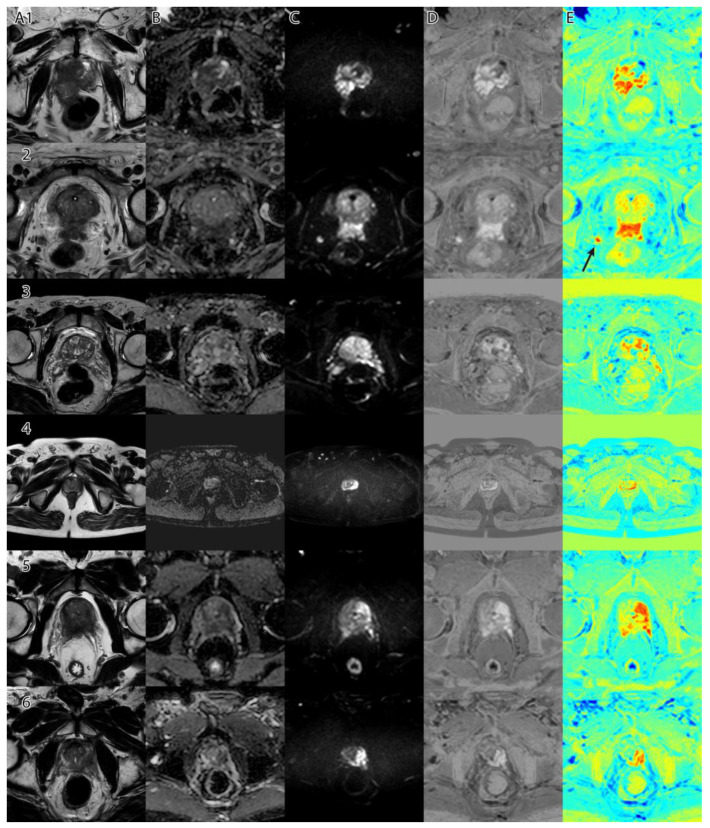
Collage of several cases and the resulting tumor intensity mask: (**column A**)—T2WI; (**column B**)—ADC; (**column C**)—DWI; (**column D**)—tumor intensity mask (grayscale); (**column E**)—tumor intensity mask with ‘jet’ color map. **E2**—Arrow shows a high risk of extraprostatic invasion of nodules.

**Figure 7 life-13-00910-f007:**
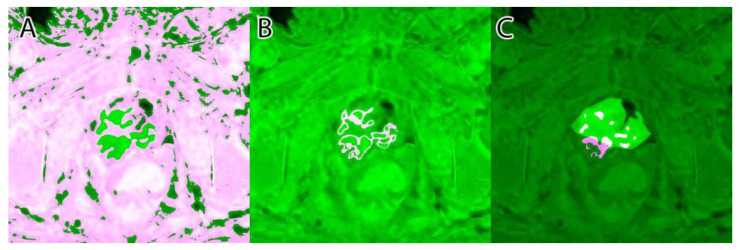
Example of overlap between the PI-RADS class intervals and the original dataset: (**A**)—no PI-RADS 3; (**B**)—no PI-RADS 4; (**C**)—PI-RADS 5 overlaps intensity mask. The remaining intensity mask from (**C**), which does not overlap with the radiologists’ annotation, is represented by the prostatic invasion of the surrounding tissue.

**Figure 8 life-13-00910-f008:**
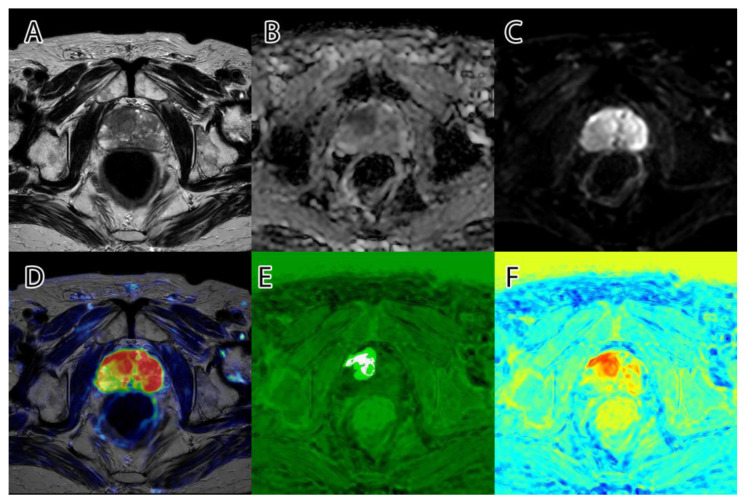
T2 shine-through effect: (**A**)—T2WI; (**B**)—ADC; (**C**)—DWI; (**D**)—T2 and DWI fused image; (**E**)—the radiologists’ annotation overlapped with intensity mask; (**F**)—our ‘jet’-mapped tumor intensity image.

**Figure 9 life-13-00910-f009:**
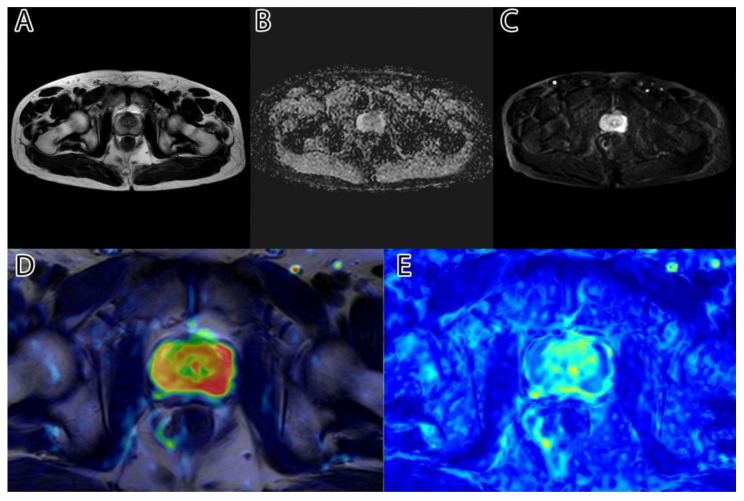
An incidental finding in a patient with extrapelvic pathology, so the field of view is large, and the protocol is not ideal. However, our algorithm proved accurate because the hypersignal area in DWI is not PCa: (**A**)—T2WI; (**B**)—ADC; (**C**)—DWI; (**D**)—T2 and DWI fused image; (**E**)— our ‘jet’-mapped tumor intensity image.

**Figure 10 life-13-00910-f010:**
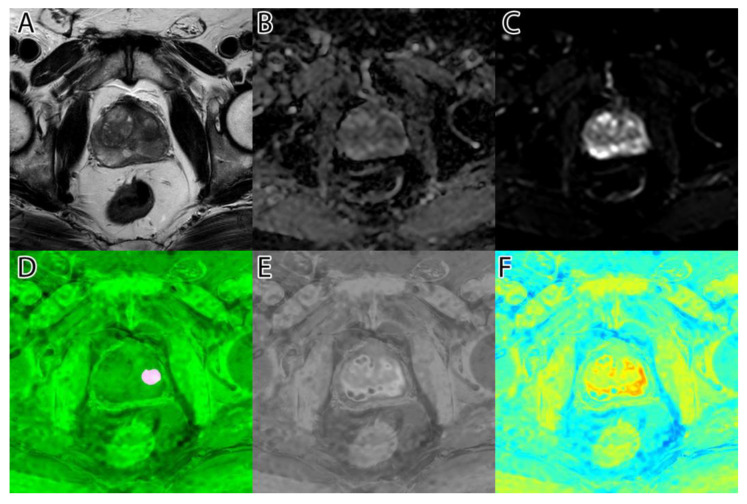
A PI-RADS 3 lesion in the posterior transition zone on the left side surrounded by insignificant areas of diffusion restriction. PI-RADS 3 lesions are only visible on T2WI, so fusion with DWI and ADC does not highlight them: (**A**)—T2WI; (**B**)—ADC; (**C**)—DWI; (**D**)—the radiologists’ annotation; (**E**)—grayscale tumor intensity image; (**F**)—our ‘jet’-mapped tumor intensity image.

**Table 1 life-13-00910-t001:** General characteristics about the sequences used in the MRI protocol for the images combined by our model.

Sequence	Matrix Size	Slice Thickness	Slice Distance	Avg. Slices
T2WI	480 × 480	3 mm or less	Same as thickness	8–16 slices
DWI	96 × 96	Same as T2WI	Same as T2WI	Same as T2WI
ADC	96 × 96	Same as DWI	Same as DWI	Same as DWI

## Data Availability

The data presented in this study are available on request from the corresponding author. The data are not publicly available due to privacy.

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
