# Peer review of "Tumor Area Highlighting Using T2WI, ADC Map, and DWI Sequence Fusion on bpMRI Images for Better Prostate Cancer Diagnosis"

_life, 2023, doi:10.3390/life13040910_

Round 1
Author Response
Thank you for reviewing my paper. Thank you even more for the fact that you read carefully and understood the work that we put in to achieve it. The encouraging words regarding the integration of such a model in clinical practice make me happy and motivate me, as a young PhD student, to continue this research further.
Below I will list each change made as a result of the issues you found or why I believe they are not a real issue:
- “The relative merits of the T2w, DWI, and ADC MRI sequences in terms of their utility are described on pages 2 and 3. However, there needs to a more precise description of how the ADC maps are calculated than is presented in lines 61-66 . ADC maps are calculated maps derived from fitting the pixel by pixel signal intensities of diffusion weighted images acquired at least two different b-values to a mathematical model describing the variation of signal intensity with increasing b-value. These models range in complexity from simple mon-exponential decay to more complicated models that include perfusion and other features. The precise model used, including the b-values employed, should be provided.”
I added details. Unfortunately, I do not have access to several aspects related to the algorithms used by those behind the platform. The most important thing I wanted to express was that the exponential ADC map was not used, otherwise the overlay with the DWI sequence would have been useless or even wrong. If you think more details are needed then I can contact the software company.
- “Using both DWI images and ADC maps in the algorithm implies that the information contained in these images are relatively independent. This can and should be shown statistically before combining the information provided by these two kinds of MR images.”
Our model follows the principles of the PI-RADS score, and its testing and verification is not the subject of our study. We proved that the information contained in DWI images and ADC maps is different (but not independent) through the discussions related to figures 8, 9 and 10.
- “Page 4 describes the fusion technique and methods for combining the information from the MR images. The authors should restate “overlapping of the two sequences” on page 143 and elsewhere in the manuscript to “combining images acquired from two different MR sequences or modalities”, since it is the data from mages that are being combined.”
I agree with this observation. I made the necessary changes.
- “Page 5 describes the role of the radiologist in segmenting each images acquired. It is likely that accuracy of this segmentation process depends on the experience and training of the radiologist. On page 12 sentence 344 is incomplete. Also the use of manual segmentation should be listed as a potential limitation of this study.”
I agree with these observations. I made the necessary changes.
- “Line 14 of the abstract: “veracity” should be replaced by accuracy or an equivalent term.”
I agree with this observation. I made the necessary changes.
- “Line 80 page 2: “the DWI sequence contains” should revised to “Images acquired with the DWI sequence””
I agree with this observation. I made the necessary changes.
- “A label of the x-axis for the histograms shown in Figure 4 should be provided. Also these histograms appear to show bimodal distributions. This should be discussed in the results section.”
I agree with this observation. I made the necessary changes.
In addition to the reported issues, I have added to this paper an example of a PI-RADS 2 patient - control subject, a table with quick specifications on the sequences used, and labeled the figures accordingly.
Other changes include small title changes: from mpMRI to bpMRI (since we do not use DCE) and from ADC to ADC map (to clarify the origin of the abbreviation).
Reviewer 2 Report
Thank you for the opportunity to review the manuscript entitled, " Tumor area highlighting using T2WI, ADC, and DWI sequences fusion on mpMRI images for better prostate cancer diagnosis."
1. It is unclear from the abstract what the ground truth is. Is it PIRADS 4/5? Biopsy results?
2. The authors mention they used artificial intelligence but don’t describe the model, including architecture, train and test set in the abstract. Please develop on this.
3. Rather than describing the different sequences in depth in the introduction, it would benefit the paper if the authors spent time describing similar studies, their pros, their cons, study samples, etc. and show how this study fares compared to those
4. Please include a table describing characteristics of the different sequences, DWI, ADC, and T2, in regards to matrix size, average number of slices in the z-axis, scanner types, etc.
5. Please avoid too much discussion type sentences in the methods section, including lines 142-152 and keep it as objective as possible.
6. I feel the papers left out two important papers. It would strongly benefit the paper if the authors cited them and discussed them as alternatives to this approach. The first is by Bhattcharya et al. where they developed a model trained on MRI and verified on whole mount histology. The second is by Soerensen et al and showed how artificial intelligence has a role and can be implemented in the clinic. Links:
Paper 1: https://www.sciencedirect.com/science/article/pii/S1361841521003339
Paper 2:
https://www.auajournals.org/doi/full/10.1097/JU.0000000000001783
7. It is a big issue that PIRADS is being used as ground truth labels and not Gleason grading from biopsy or whole mount histology. Please use this as the first limitation and address it properly
8. Please specify b values for DWIs
9. Please provide a description for Figure 4
10. Also label Figures 5-9
Author Response
Thank you for reviewing my paper. Surely the issues you found were different from those of the other reviewer. Most started from the premise that this is an artificial intelligence model (partially due to us because we left these words in the conclusions by mistake) but there were other parts of the text that mentioned that our approach was different.
Below I will list each change made as a result of the issues you found or why I believe they are not a real issue:
- „1. It is unclear from the abstract what the ground truth is. Is it PIRADS 4/5? Biopsy results?”
I agree with this observation. I made the necessary changes. The ground truth is the segmentation of lesions (PI-RADS 3, 4 or 5) marked by experienced radiologists, the purpose of this model being to mark similar lesions.
- "2. The authors mention they used artificial intelligence but don’t describe the model, including architecture, train and test set in the abstract. Please develop on this.”
We mistakenly mentioned (in the conclusions) that we used artificial intelligence. I have corrected the mistake. In the rest of the paper, we explain that our model combines images from bpMRI sequences and automatically marks suspicious lesions based on the principles of the PI-RADS score, without using any deep learning method.
- „3. Rather than describing the different sequences in depth in the introduction, it would benefit the paper if the authors spent time describing similar studies, their pros, their cons, study samples, etc. and show how this study fares compared to those”
I didn't deleted anything from the introduction because the reviewer really appreciated this section of the paper. We have described and compared similar studies in discussions – lines 346-380.
- „4. Please include a table describing characteristics of the different sequences, DWI, ADC, and T2, in regards to matrix size, average number of slices in the z-axis, scanner types, etc. „
I agree with this observation but I don't understand what you mean by scanner types. All images were taken by the same Philips Ingenia 3.0T MRI scanner mentioned in the dataset. I added the table to the MRI protocol.
- „5. Please avoid too much discussion type sentences in the methods section, including lines 142-152 and keep it as objective as possible.”
I agree with this observation. I believe that the place of lines 142-152 is in the discussion section, so I moved them.
- „6. I feel the papers left out two important papers. It would strongly benefit the paper if the authors cited them and discussed them as alternatives to this approach. The first is by Bhattcharya et al. where they developed a model trained on MRI and verified on whole mount histology. The second is by Soerensen et al and showed how artificial intelligence has a role and can be implemented in the clinic. Links:
Paper 1: https://www.sciencedirect.com/science/article/pii/S1361841521003339
Paper 2: https://www.auajournals.org/doi/full/10.1097/JU.0000000000001783”
Since our model does not use machine learning methodology, it is very difficult for me to make a direct comparison between them. However, I think that these two articles are really impressive and I cited them along with the other models that caught my attention.
- „7. It is a big issue that PIRADS is being used as ground truth labels and not Gleason grading from biopsy or whole mount histology. Please use this as the first limitation and address it properly”
I do not consider this to be a limitation of the study. Our model does not claim to be able to identify lesions more efficiently than by applying the PI-RADS score. It would certainly have brought an advantage to compare the lesions described with the histological interpretation, but the result would have been similar to the specificity of PI-RADS, where there are many studies related to this.
- „8. Please specify b values for DWIs”
I agree with this observation. I added the information in the introduction, as well as additional data about the ADC reconstruction used.
- „9. Please provide a description for Figure 4”
I agree with this observation. I made the necessary changes.
- „10. Also label Figures 5-9”
I agree with this observation. I labeled all the figures accordingly.
In addition to the reported issues, I added to this paper an example of a PI-RADS 2 patient - control subject, made small necessary changes to figure 4 (now figure 5), and described it in more detail.
Other changes include small title changes: from mpMRI to bpMRI (since we do not use DCE) and from ADC to ADC map (to clarify the origin of the abbreviation).
Round 2
Reviewer 2 Report
The authors did a great job accessing comments. No further comments.